# Business Ethics Crisis and Social Sustainability. The Case of the Product "Pura Vida" in Peru

**Renato Peñaflor-Guerra** [1], **M. Victoria Sanagustín-Fons** [2,*]  **and Julianna Ramírez-Lozano** [3]

1   Communication and Corporate Image Program, School of Communications, Universidad Peruana de Ciencias Aplicadas, Lima 15023, Peru; renato.penaflor@upc.edu.pe
2   Department of Psychology and Sociology, University of Zaragoza, 50009 Zaragoza, Spain
3   CENTRUM Católica Graduate Business School, Pontificia Universidad Católica del Perú, Lima 15023, Peru; julianna.ramirez@pucp.edu.pe
*   Correspondence: vitico@unizar.es; Tel.: +34-653-13-34-99

**Abstract:** Peru is a Latin American country with one of the most powerful and dynamic emerging economies in the world; however, it reveals considerable inconsistencies and socioeconomic inequalities. This research demonstrates that business ethics and corporate social responsibility are closely related with the welfare state of the country/region where a company is incorporated. Research work has been carried out on the case of the product "Pura Vida", of the Gloria Company, implementing a mixed research method (documentary, *quan* and *qual*) in which a descriptive collection of data from different sources, in relation to the company and the crisis with *Pura Vida* milk, has been used; additionally, an *ad hoc* survey has been conducted on a sample of Peruvian citizens to know their insight in relation to the relevant aspects of business ethics and corporate responsibility and their opinion regarding the specific case being investigated; finally, in-depth interviews were held with the company's management staff. It is shown that without a certain degree of welfare state, it is difficult to implement ethics and social responsibility in companies and in society as a whole. In addition, the main concerns of Peruvians regarding ethics and social responsibility can be observed.

**Keywords:** business ethics; corporate social responsibility; Peru; *Pura Vida*

## 1. Introduction

Peru is one of the countries with the most dynamic emerging economies in Latin America, and this has been the case during the last ten years. In 2017 the country's GDP reached 211.4 billion US dollars [1]; however, the country is going through a socioeconomic situation full of contrasts, characterized by numerous differences, imbalances and social inequalities. The social policies the government implemented to combat poverty and inequality are not efficient, mostly based on "orthodox" policies promoting foreign investment in the primary export sector (mining, oil, fishing, soybean cultivation, etc.), which was considered the main driver of growth, and conservative fiscal and monetary policies that created a climate of confidence for investors and led to stable exchange rates and prices [2]. This circumstance is paired with problems of various kinds, not only economic but also cultural, educational and regarding the empowerment of citizens, and therefore of inhabitants and potential consumers. Moreover, Peru is seen as one of the countries with the highest social perception of corruption, according to data from the Transparency International Organization (2018) [3]. Currently, the five former presidents of the republic were sentenced and investigated for corruption offenses. Following the classic work of Karl Polanyi (1957) [4,5], the market must be limited by laws and certain types of regulations, otherwise it may negatively influence society, but this is always under the idea of a correct government and adequate public policies. Corporate Social Responsibility

(CSR) refers to the impact of an enterprise on the economic, environmental and social spheres in which it operates [6,7]. CSR can either be regarded as pursuing objectives similar to the welfare state (reducing the negative effects of market forces) or as antagonistic to these objectives (furthering of market forces) [8,9]. In this context, the ethical business commitment is really difficult, and the CSR strategies and policies applied by companies are limited, although there are certain exceptions worth being described and investigated [10]. This has changed governments' capacity to act on social and environmental issues in their relationship with companies, but has also affected the framework in which CSR public policies are designed: governments are incorporating multistakeholder strategies [11]. Moreover, some NGOs and social movements push CSR as an instrument for stimulating economic equalitarian social opportunities [9]. CSR is applied differently depending on the countries, welfare states and cultural and political environments, accentuating how the nuanced forms of CSR in the developing world are invariably contextualized and locally shaped by multilevel factors and actors embedded within wider formal and informal governance systems [12].

Although the social perception of the ethical problem and the unethical behavior of companies is increasing, a meaningful new force is still needed. In our research, we noted how the environmental commitment and the organizational justice toward the workers, the fair relationship with suppliers and, in general, the position of being a company concerned with the environment and the people who work in it, has made the perception and concern about these issues increase in Peru. Similarly, the research made by Lopatta, Jaeschke and Chen (2017) [13] shows that there is a positive relation between state-controlled ownership and the CSR performance of firms, whereas the other types of controlling ownership have no impact on CSR performance. Further results show that the evidence is more pronounced in countries with more stakeholder engagement.

Under the umbrella of ethics and social responsibility, a relevant concept emerges in this research: the so-called Corporate Ethical Discomfort/Welfare, a term coined by the authors and which fits perfectly in the present study, based on the complaints made by the Peruvian media, stressing the lack of commitment of the company Gloria and its product "Pura Vida". We understand the term Corporate Ethical Discomfort/Welfare as a dichotomous concept that refers to the permanent ambivalence certain companies present in emerging economies; furthermore, reaching a certain ethical and responsible commitment implies well-being (everyone wins: the company, the workers and society). But that is blinded by the exclusively economic benefit approach that continues to weigh on companies like Gloria.

Authors such as Prahalad and Hamel (1990) [14] explain the concept of the base of the pyramid (BOP), and were among the first to talk about the market potential at the base of the pyramid (BOP). By focusing on the unmet needs of low-income populations, companies can create profitable markets and, at the same time, help the poor address some of their most urgent requirements. But this vision is not fully shared by companies in emerging economies. To assume this concept, companies must have a culture of social innovation (CSI); this term was introduced in 1999 by Rosabeth Moss Kanter of Harvard Business School [15]. Companies should use social problems as a learning laboratory in order to identify unmet needs and to develop solutions that create new markets. Meanwhile, few companies and organizations have a socially responsible behavior, and they do so thanks to the leadership and ethical culture of the founder and managers. What are the reasons for promoting this hypothesis? Principally, the fact that it shows how society is limited to encouraging and reinforcing the correct behaviors of organizations and companies, mainly because the determining factor that moves the consumer/citizen is the price and not others type of value; thus, the first consequence is the acceptance of social performance, given that consumers/citizens are trapped in ineffectiveness and therefore do not report due to lack of both economic and cognitive resources. Our theoretical conclusion is that the environment of the company, together with social attitudes, determine the company's behavior and commitment to relevant social values such as welfare, transparency, loyalty or the so-called accordance with the logic of the dominant government. Enterprises are more or less responsible depending on the social response they receive. Above all, we consider the welfare state as the main socioeconomic requirement to achieve the ethical and social responsibility objectives of the company.

Corporate Social Responsibility (CSR) can be considered as a global paradigm [16], and the content of CSR has evolved over time, depending on historical, cultural, political and socio-economic drivers, and on the particular conditions of different countries at different points in time [17]; it has become a central theme of the political agenda of many companies since the second half of the last century. Some authors, as Sulkowski et al. (2008) [18], find that the impacts of culture and political economies on what societies believe, expect and say about CSR are important. Despite these cultural differences, CR has an important role inside each company. Specifically, they studied companies from from China, India and Japan. Other authors, such as Panapanaan (2006) [19], say that the Finnish development and the welfare state system, globalization, stakeholders and the pursuit of sustainable development are the main drivers of CSR. In this sense, Midttun (2005) [20] has developed an emerging model of corporate social responsibility (CSR), or embedded relational governance, which seems to share the basic market orientation of the liberal model, albeit at the same time sharing many of the social and collective goals of the welfare state.

According to the EU's definition (2018) [21]: "Corporate social responsibility is the voluntary integration of social and ecological aspects in everyday corporate operations with the environment." This approach is connected with the basic principles of sustainable development [22]: economic, environmental and social. CSR is considered a new paradigm which helps to think holistically and systematically about the main aspects of a company in relation not only to the people who make it up, but also to the society in which the company is integrated, the role of the company with regard to development, the way in which business is carried out (internal and external), corporate governance, poverty alleviation, the corporate contribution to peace, the fight against terrorism and, finally, the association between companies, government and civil society: a common basis and a collective action (based on minimum values). However, there are specific barriers to applying CSR in the strategy and political management of the company (usually voluntary), as can be seen in Table 1. Wilson (2012) [23], to overcome voluntarism, proposes a regulatory support for the development and growth of social enterprises, such as community development finance institutions. The aim is to overcome vulnerability and unequal opportunity with respect to financial services, for example.

**Table 1.** Barriers to the implementation of Corporate Social Responsibility.

| At Business Level | At Country/Society Level |
|---|---|
| Lack of leadership and vision | Lack of creative pressure from government and civil society. |
| Excessive attention to short-term objectives | Lack of consumer support. |
| Inability to recognize opportunities | Lack of peer support through trade associations: reluctance of other companies to follow |
| Lack of entrepreneurial spirit and innovation. | Lack of economic/market incentives |

Source: [24].

The classic definition of the four dimensions of Carroll and Bucholz (2014) [25], which includes the understanding of the four components of CSR—economic, legal, ethical and discretionary (philanthropic)—is currently being examined due to different business scandals and political corruption (see Table 2).

Based on the consideration of these components in relation to CSR, we have analyzed and investigated the importance of this paradigm in organizational behavior today. We expand further to establish, on the one hand, that the minimum requirement to move forward with ethics and corporate social responsibility is the welfare state of a society, and to determine, on the other hand, the social perception of these aspects [26], in order to understand how and why people behave in a certain way against those companies that have been acting improperly and have even been denounced [27,28].

**Table 2.** Definition of the four dimensions of Carroll.

| Responsibility | Social Expectation | Examples |
|---|---|---|
| Economic | Required | Cost effectiveness. Maximize sales and minimize costs. |
| Legal | Required | Obedience to laws and regulations |
| Ethics | Expectation | Do what is right, fair and reasonable |
| Discretionary (Philanthropic) | Desired/Expectation | Be a good corporate citizen |

Source: Carroll and Bucholz, 2014 [25].

Likewise, business ethics as a theoretical paradigm (in the Khunian sense of the term) aiming to explain the business phenomenon and develop it through the keys of ethical behavior, is closely linked to the paradigm of company culture (organizational culture). Which means discovering the existence of a series of values [29–31], so that the company becomes a community and the people integrating it can find psychological, emotional and social well-being [32]. In order for the community to survive, the different values become moral standards that individuals will use as a reference in their decision making. The individuals will be determined by those standards, but will also participate, in turn, in setting them. It is a game of interactions necessary for a human group to survive [33]. The welfare state of Sweden has recently encountered some difficulties, and this puts pressure on other stakeholders, such as corporations, to step forward and take social responsibility where the public domain failed to do so [34]. Lindh (2014) [35], using Swedish survey data collected in 2011 and latent class analysis, demonstrates that most Swedes in favor of CSR are highly supportive of state intervention in the market, and finally concludes that Swedes continue to think of public authorities as the ultimate institutional guarantor of social welfare.

The present research aims to determine whether this theoretical paradigm is incorporated in Peruvian companies. As we shall see, the appearance of certain social values determines not only the need for internal procedures, but also the outlining of the so-called external image of the company [36,37]. Both aspects contribute to business competitiveness and its survival—or at least, as pointed out by Cortina (2002) [38], despite the uncertainty of the environment, excellent companies and ethical companies increase their competitiveness in a Darwinian market [39].

It is an undoubted commercial advantage that nowadays social groups receiving the influence of a company exert some pressure on it at the same time (stakeholders' theory), responding to the expectations and demands of these groups and not only to shareholders [40]. The coordinate lines between the ethical and the economic meet at one point, in which certain social achievements are obtained through economic efficiency, as shown in Figure 1.

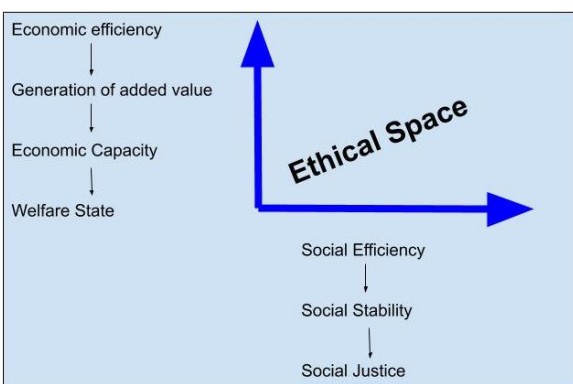

**Figure 1.** Components of the ethical space. Source: partially adapted from García Echevarría (2016) [41].

The language of ethics leads us to consider things from a perspective that other approaches overlook. For instance: Is the company responsible to others? The language of responsibility is only part of the language of ethics. For ethics to be effective in organizations and to prevent people from exploiting it for their own benefit, it is necessary to preserve the denotative meaning of ethical language (denotative reference to actions and attitudes/connotative motivational meaning). It is not only a matter of principles but also of consequences [25]. Ethics implies that people have the freedom and power to respond, that is, the responsibility and power to consider different opinions, to analyze the strengths and weaknesses of the options and to make choices based on the merits of each alternative. An ethical perspective focuses on action and not on behavior, looking for reasons to justify actions instead of explaining behaviors and recognizing the difference between "must be" and "being".

This means that people behave according to what they have decided is right. When we ask about the reasons for the action, we seek justifications and not explanations. Ethics justify the action instead of explaining the behavior. The justification shows what principles or values the actor uses when making his choice and behaving in a certain way. An ethical perspective considers that others are individuals with their own power and will. From an ethical perspective, the perception of the moral capacity of the staff, of their power and freedom to choose what is right, may well mean that the management and middle management of companies, including the entrepreneurs themselves, will choose to include the collaborators in decision-making processes.

Information and knowledge lead in the beginning of the new millennium; Castells (2002) [42] makes this clear in his book "The age of information". Regardless of the scope of our activity, and as beings proactive toward reality, if we want to survive, we must handle information. Besides, there is a change or mutation in the cognitive parameters that guide business management, that can be summarized in Table 3.

**Table 3.** Change Parameters from modern to postmodern companies.

| Mutational Parameter | Modern Company | Postmodern Company |
|---|---|---|
| 1st Organizational | Systems and procedures | Culture and behaviour |
| 2nd Relational | Communicative emission | Reception of the communication |
| 3rd Motivational | Desiderative trends | Effusive trends |
| 4th Ethics | The best for the greatest number of people | Beneficiaries and quality of goods |
| 5th Cognitive | Emphasis on procedures | Emphasis on virtues |

Source: Partially adapted from Llano-Cifuentes (1994) [43].

The change is evident both in the adaptive responses of companies through their management methods and in the different paradigms or comprehensive models of the business phenomenon. Why is it necessary to point out this circumstance? Precisely because the social and human sciences applied to the company review show an evolution aiming to the humanization of work and business, an evolution that coincides with the integral development of people. Moreover, civil society today will hardly accept business management that is driven more by technical-organizational issues than by people management. In Western countries, European citizens have assimilated the privileges of the welfare state in such a way that business management cannot ignore this fact. Our main research questions are: What are the main concerns of Peruvians regarding ethics and corporate social responsibility? How do Peruvians respond to a case in which the company's ethics and social responsibility are compromised, as in the case of "Pura Vida", that has been publicly and legally denounced? How do companies respond in an emerging economy with a weak welfare state?

## 2. Materials and Methods

The main hypothesis of our research is that a company's commitment to ethics and social responsibility is only possible if there is a general and basic welfare state in the society where the

company operates; otherwise, the company's lack of ethics is not perceived socially in a such a radical way that the company is forced to divert its business strategy. Moreover, these issues are not valued as important for society as a whole, observing certain subtle differences depending on the socioeconomic status and sociocultural level of the people, as shown in the empirical data of our research. Thus, it can be noticed that price is the most important determinant of the consumer's behavior, as revealed by the data. Table 4 shows the main elements of our research.

**Table 4.** Strategy of the case study.

| | Social Research Techniques | Database | Research Objective | Empirical Perspective | Temporality |
|---|---|---|---|---|---|
| Methodological approach. Phase Ia. | Documentary analysis | -Bibliographic data -Economic reports | -General theoretical structure -Economic situation of the company | Descriptive | Last 5 years |
| Methodological approach. Phase Ib. | Content analysis of the mass media | News in the mass media | Ethical problem | Descriptive | Last 5 years |
| Methodological approach. Phase II. | Secondary data analysis | Official databases | State of Peruvian Welfare | Deductive | Cross-cutting approach |
| Methodological approach. Phase III. | Survey | Own | Peruvian Sample | Quantitative | July, August and September, 2018 |
| Methodological approach. Phase IV | In-depth interviews | Own | CEO of the company and legal representative | Qualitative/ Inductive | September, 2018 |

Source: Own elaboration.

In order to refute or verify our hypotheses, a case study research has been developed following four methodological stages and phases. First, the documentary analysis has had a double orientation: the theoretical framework of the investigation and a description of the Gloria Company as a family of traditional Peruvian SMEs and their financial statement during the last five years. The public and legal complaint about one of its most emblematic product, "Pura Vida" milk, is also described through the content analysis of media reports. Secondly, for our research questions and hypotheses, secondary data related to the welfare state of Peru and some important official data connected to this emerging economy are shown and analyzed.

Thirdly, a random survey focused on the social perception of ethics and corporate social responsibility was carried out on a sample of five hundred people. The survey assesses the impact level of the business crises on the perception/impact of the image and reputation of the product brand and corporate brand. Reputation variables were analyzed: product quality, business leadership, social responsibility. Likert scales were used for the questions, which were oriented toward the assessment of the attitude of Peruvians in relation to all these aspects. Moreover, we challenged their possible behavior in this specific case of company ethics. Finally, an in-depth interview was held with two representatives in charge of the company, the social responsibility manager and the lawyer of the company, who shared made their views on the ethical issue and the strategies aimed at solving the problem.

## 3. Results

The obtained results are analyzed following a scheme including a descriptive collection of the relevant data in relation to the Gloria company and the "Pura Vida" case, a basic statistical analysis of the survey content and finally an intuitive hermeneutic interpretation by the researchers of the interviews held with those responsible for the company.

### 3.1. History of the Gloria Company and the Crisis of Pura Vida

The company Gloria S.A., known today as Leche Gloria S.A., is a family business founded in 1942, and has been the market leader in dairy products during the last five decades in most of the country. Most Peruvians consume their milk and dairy products. In 2017, the company was reported in relation to one of its main products, consumed by millions of Peruvians, called "Pura Vida" milk, bearing a label with the name "milk". However, this product was not milk, but a dairy mixture that did not have the percentage of cow's milk to be named that way. Table 5 shows the description of this case, with the main identifying features like the type of company, leadership, complaint, media role, social perception and the business strategy carried out to face the situation, that dominated the various mass media of Peru for a while.

**Table 5.** Description of the case study.

| | Type of Company | Type of Leadership | Evolution of Economic Growth during the Last 5 Years | Reporting Year of the Business Ethics Problem | Role of the Mass Media | Social Perception | Business Strategy |
|---|---|---|---|---|---|---|---|
| The "Pura Vida" case of the Gloria Company in Peru | Family SMEs | Pyramidal and Patriarchal | Growing, emerging economy | May, 2017 | Report and media attention to the problem | See the analysis of the results and the conclusion | Marketing campaigns and other audiovisual strategies |

Source: Own elaboration.

It is important to highlight that the complaint was made in Panama, since this product was being sold there. Later, when the news arrived to Peru, the authorities revalued the product and the labeling authorization, deciding to prohibit its consumption and requesting its withdrawal from the market. Three months later, the product was reintroduced to the market under the name of milk mixture, with the image of the cow removed from the label. To date, the product is still on sale and being purchased and consumed in the same proportion as before the crisis.

The Gloria Group is an industrial conglomerate of Peruvian capitals with businesses in Peru, as well as in Bolivia, Colombia, Ecuador, Argentina and Puerto Rico. Their activities are mostly in the dairy and food sectors, but they also have businesses in other sectors, such as cement, paper, agribusiness, transportation and services. It should be noted that Gloria is a seventy-year-old Peruvian brand, well known and respected for its dairy products, among which the evaporated milk "Leche Gloria" stands out. It has a representation in the dairy market of more than 52%, followed by competing companies Laive (15%), Ideal (6%), Bonlé (3%) and Nestlé (3%), according to a study of the Arellano company (2018). The company has also been awarded several times as one of the 100 best companies in emerging markets, according to the Global Challengers 2016 list (The Boston Consulting Group, Boston, MA, USA, 2016). Gloria has also been recognized as a Top of Mind brand from Peru, since its dairy products are one of the most used since childhood, as they generate trust and a positive experience for the user.

According to a study carried out by the independent researcher Ernesto Linares about the families with the greatest assets in Peru, the owners of the Gloria Company stand out, occupying the third place in the list. The Rodríguez Pastor family, with its two top representatives, Vito and Jorge Rodríguez, is one of the richest families in the country. This family group has an estimated wealth of US $4800 million.

As mentioned, the Gloria Group is known in Peru due to the production and commercialization of dairy and food products such as evaporated milk, fresh milk, yogurt, cheese, butter, margarine, juice, etc., which are sold under the umbrella brand Gloria. However, they also have some products with their own brands, such as "Pura Vida" milk.

The events surrounding this crisis are presented chronologically as follows:

- 2 June 2017: the media and networks reported that AUPSA (Panamanian Food Safety Authority) had concluded that the "Pura Vida" product, produced and commercialized by the company

Gloria S.A., could no longer enter the country under the label of "milk", as it was made up of 40% of elements other than those of natural milk. In addition, the withdrawal of the product from the market was ordered.

- 4 June 2017: Gloria Company issued a first statement.
- 6 June 2017: The Peruvian Association of Consumers and Users (ASPEC) issued a statement saying that "Pura Vida" and "Reina del Campo" were not products that could be marketed as "evaporated milk", because that contravenes the National Technical Standard and the *Codex Alimentarius*. It was also pointed out that these products must come from a process of partial water elimination, so that their composition shares milk characteristics, and that presenting the image of a cow on the label misled the consumer.

For its part, the Ministry of Health of Peru issued its first statement, saying that the General Directorate of Environmental Health and Food Safety (DIGESA) had granted some sanitary registries with the name of "Evaporated dairy food" in 2014 and that a series of modifications were made to name the product as "partially skimmed evaporated milk with soy milk, malt dextrin, vegetable fat and minerals (iron and zinc) and enriched with vitamins A and D" in 2015. The company Grupo Gloria S.A. issued a second statement reaffirming its commitment to consumers.

- 7 June 2017: The Ombudsman, Walter Gutiérrez, of the Ombudsman´s Office of Peru, said: "The information presented on the label is highly misleading. It violates two rules: duty of information and suitability. They frustrate the customer´s expectations". Simultaneously, Gloria S.A. issued a third statement in line with the previous ones.
- 8 June 2017: DIGESA (General Directorate of Food Health of the Ministry of Health of Peru) stated that it granted the permit in 2014 at the request of agribusinesses. About this matter, DIGESA said: "We feel outraged, we know that it is not milk ('Pura Vida'), that is why we are admitting the mistake and we will correct it" (María Eugenia Nieva, general manager of the Directorate of Authorizations and Certifications of DIGESA, the entity responsible for issuing sanitary permits). Meanwhile, Indecopi, a consumer protection regulatory agency in Peru, ordered Grupo Gloria to stop the sale of "Pura Vida" Nutrimax throughout the country.
- 9 June 2017: An interview to the general manager of the Gloria Group, Robert Priday, was published in the newspaper El Comercio, in which he emphasized: "I think that Peruvians will forgive us". He also pointed out that the company had always complied with the regulations and that they had acted according to the law. Therefore, they should not have penalties or fines. Moreover, the Ministry of Health, through DIGESA, issued a second statement.
- 27 June 2017: Seven officials from DIGESA were investigated for the "Pura Vida" case. Two are no longer working at DIGESA and five have been expelled from their positions. That day, Gloria S.A. issued its fourth statement, denying any kind of scam or deception toward consumers.

Faced with such a situation, the company developed a communication strategy. At the internal communication level: there were internal emails inviting the company's employees to be direct participants of an image recovery campaign called "I am Gloria". Messages with emotional content regarding the affection that Gloria workers had for the company were sent by mail and multiplied in social networks. At the external communication level: company representatives offered interviews on the main radio, television, and written press networks of the country. They recognized that the "Pura Vida" product was not 100% cow's milk, but only 60%, the other 40% consisting of other ingredients. However, they indicated the whole time that the company had complied with the law and they never apologized to the consumers. Also, two television spots were made, with the participation of collaborators from the Lima and Arequipa plants, including middle managers and workers, giving their opinion on the damage that the crisis was causing to the image of the company. However, they did not receive very good comments, as many people mentioned that they were not credible. On a commercial level, the company was forced to change the label of the "Pura Vida" product, as well as the advertising spot. The most relevant change was on the can's label: while in the initial version there

was an image of a cow, in the new version there was no cow. Some of the most direct publications that were issued after the crisis took the form of memes, consisting of photographs in which images of the "Pura Vida" milk indicated it was a low quality product without cow's milk.

*3.2. Descriptive and Relational Statistical Analysis*

The results of the survey included 438 valid responses from Peruvian citizens coming mostly from Metropolitan Lima (Lima and Callao), of both sexes (35.8% men and 64.2% women), with an average age of 30.6 and diverse socioeducational levels: complete Secondary Education (43.8%), complete Technical/University Education (38.8%) and complete Postgraduate Education (17.4%). The survey was answered anonymously, and the main results have been analyzed using the SPSS program and setting explanatory cross-references of the phenomenon addressed.

The chosen sample perceived the Peruvian company as moderately responsible; that is, they were aware of certain power abuses by the organizations and that these had happened due to the lack of social commitment on the part of certain companies. Table 6 shows the opinion of the respondents about the Gloria Company after the complaint regarding the "Pura Vida" product. According to these data, there were no major differences in the opinion of the respondents regarding the Gloria Company after the scandal. It can be noted that the opinion varied little between a worsening of the opinion (43.4%) and the maintenance of the opinion (45.4%) about Gloria; however, the opinion on the company worsened significantly (52.6%) after the scandal and the accusation for the respondents with a university Postgraduate Education level.

**Table 6.** Has your opinion of the company producing the "Pura Vida" brand improved, remained the same or worsened after this problem?

| Last Level of Study Completed by the Respondent (RECODIFIED) Cross Tabulation | | | | |
|---|---|---|---|---|
| | Last Level of Study Completed by the Respondent (RECODIFIED) | | | Total |
| | Secondary | Technical/University | Postgraduate | |
| Worsened | 43.8% | 38.8% | 52.6% | 43.4% |
| Maintained | 43.8% | 49.4% | 40.8% | 45.4% |
| Improved | 12.5% | 11.8% | 6.6% | 11.2% |
| Total | 100.0% | 100.0% | 100.0% | 100.0% |

Source: Own elaboration.

It is significant that when questioned about the "Pura Vida" case, most of the respondents, as shown in Figure 2, knew about the problem, since it had been widely communicated in the media and reported to the competent public bodies.

Nevertheless, the majority of respondents had forgotten, erased or eliminated from their cognitive perception the concrete problem of the crisis, widely spread by the mass media—a somewhat paradoxical issue, given that the broad and significant majority had heard of the crisis affecting "Pura Vida" and the Gloria company itself.

Regarding the question: What is your opinion about the "Pura Vida" product? it can be observed in Figure 3 that a high percentage (25%) did not have an opinion in this regard, while a majority of respondents had a regular opinion of the product (32.40%), that is, they neither had a good nor a bad opinion. Finally, a minority had a very bad opinion (13.7%), while 10.50% still had a good opinion of the product.

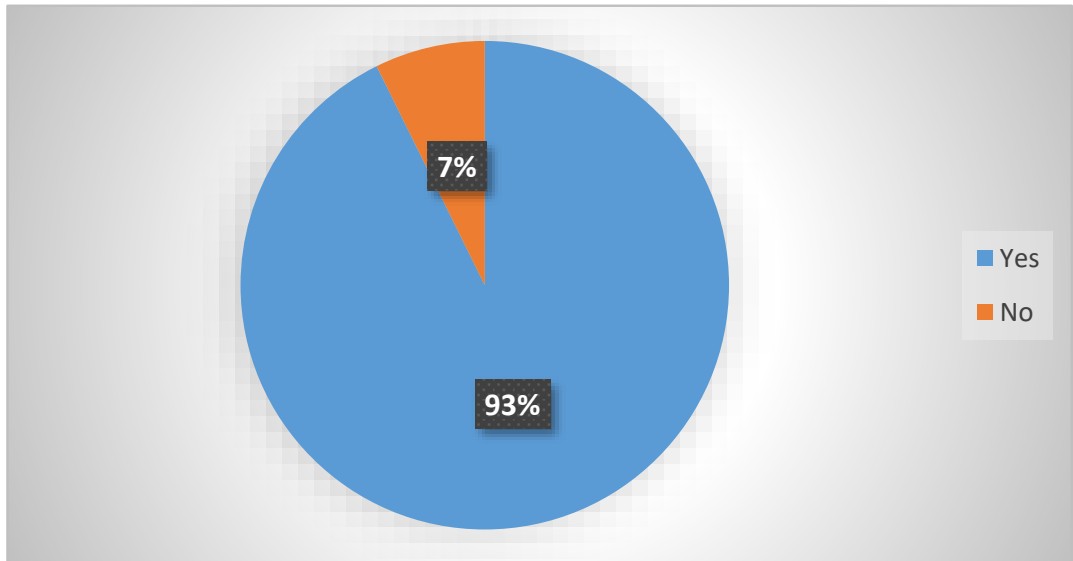

**Figure 2.** Have you heard about the problem/demand faced by the "Pura Vida" product in May 2017? Source: Own elaboration.

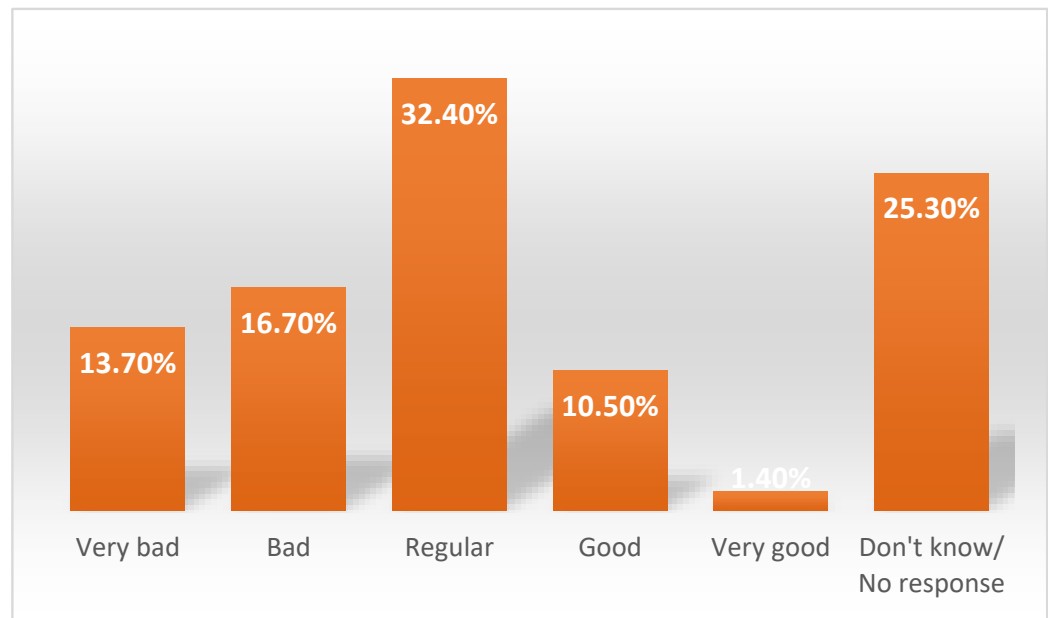

**Figure 3.** What is your opinion about the "Pura Vida" product? Source: Own elaboration.

It is interesting to highlight the aspects regarding the social responsibility of companies that mainly concern the surveyed sample. As can be seen in Figure 4, the environmental issue was considered as predominant when the sample was requested to assess the importance of this aspect of CSR. Specifically, it was considered important to adopt measures and practices toward the environmental care; the reasons might have been mainly motivated by an international social trend of evident environmental concern. On the other hand, keeping in mind the degree of pollution and problems of overload and overcrowding in a city like Lima, all of this is reflected in the responses to a survey such as the one that has been carried out.

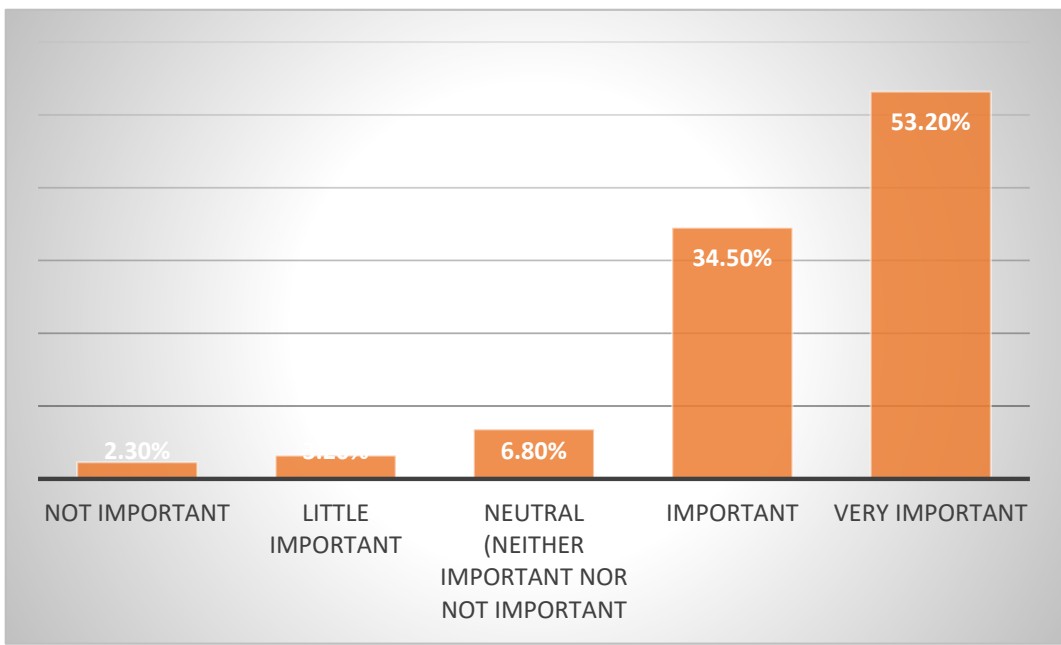

**Figure 4.** Adopting measures and practices for environmental care. Source: Own elaboration.

Guaranteeing ethics in business management was another aspect that concerned the Peruvians that were surveyed (49.80%) (Figure 5), when asked to assess the importance of this aspect of CSR. This shows that there really is a call for organizations, in this case companies, to commit themselves to moral principles of well-doing and well-being toward society as a whole. It is a way to consolidate projects, by observing minimum values with which most people agree, but whether that materializes later in the buying behavior is another issue.

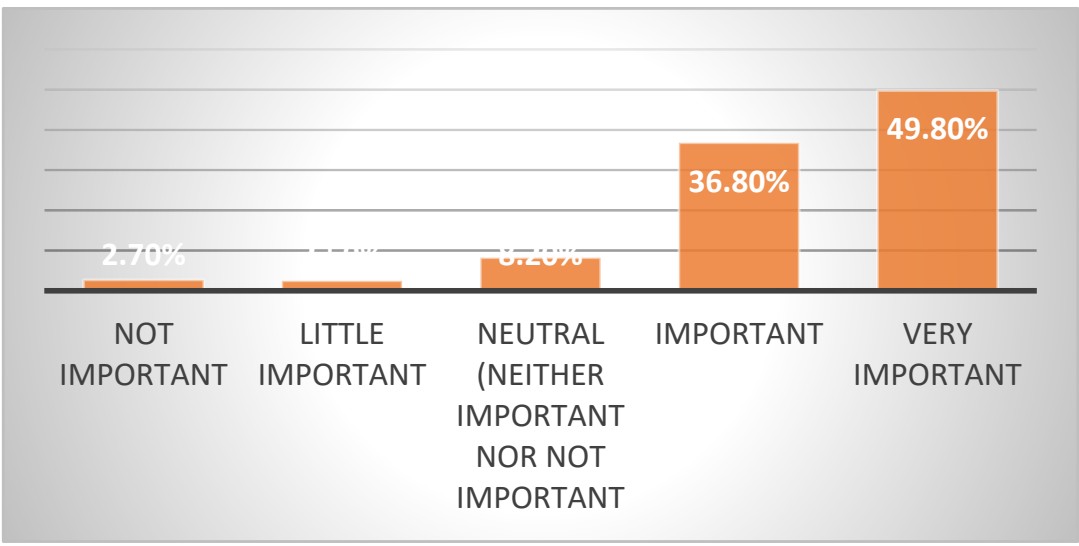

**Figure 5.** Ensuring an ethical behavior in company management. Source: Own elaboration.

Likewise, an issue that concerned a good percentage of the respondents (56.60%) was the need to comply with the law and the guidelines that regulate businesses, as revealed in Figure 6, when they were asked to indicate if they considered it interesting to ensure an ethical behavior in company management. From our perspective, this has a double meaning: on the one hand, it shows that there are degrees of misunderstood freedom in which many public and private organizations do not comply with the laws; on the other, it shows that citizens really want and demand this not to happen. There is

a kind of victimhood in relation to non-compliance with the rules by those who hold economic power and—why not say it—political power too.

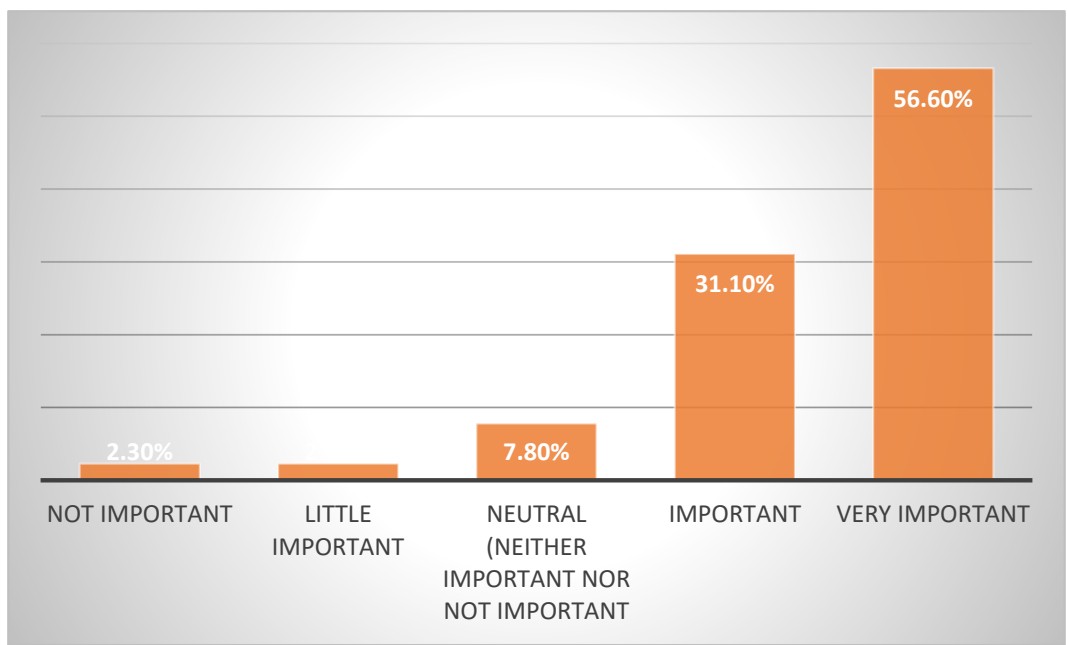

**Figure 6.** Complying with the law and guidelines that regulate the business. Source: Own elaboration.

Another aspect that most concerned respondents (52.50%) was to ensure customer rights, and that is corroborated by the data provided about ethics, compliance with laws and environmental care. Thus, an evident social trend at the global level is spreading slowly but inexorably throughout the world, including Peru. (See the approval by the United Nations (2015) of the Sustainable Development Goals). However, there is still time for this concern to become generalized, as shown in Figure 7.

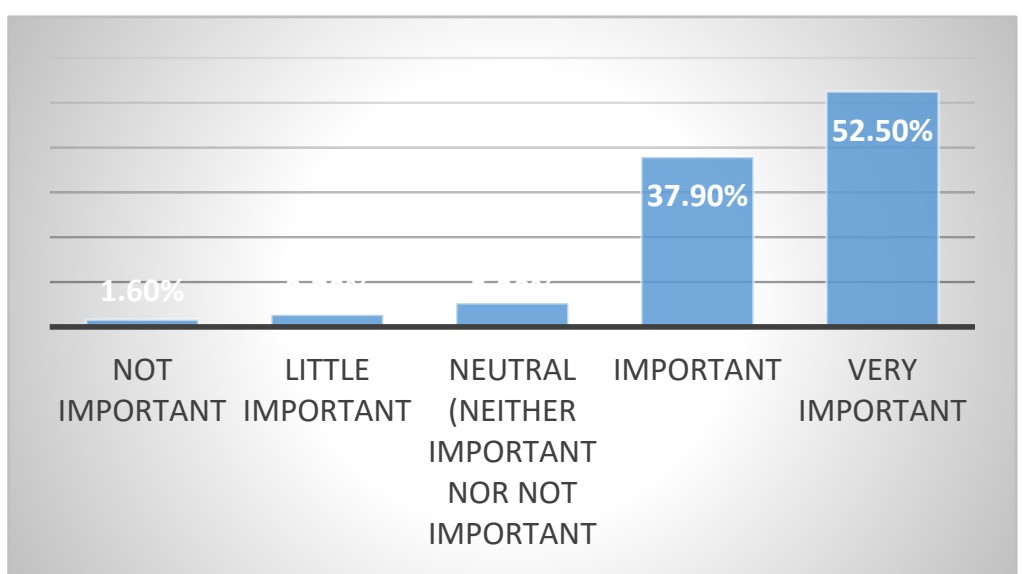

**Figure 7.** Attending to, and protecting, customers´ rights. Source: Own elaboration.

Other aspects related to CSR generated less concern, such as creating decent employment in the community, improving the employment status of workers or developing educational projects in the community, and the balance between family and work life.

As for the question of why companies are concerned with adopting social responsibility policies, the answers reveal a cultural trend that is specific to Peru, because it reflects aspects related to an external dimension that includes opinion and social control (image, publicity), an internal and even subjective dimension related to beliefs (the company believes that it will do better with these policies) and finally an altruistic motivation, such as improving society. More specifically, the majority, regardless of academic level, sex or age, believed that the most important reason was an issue of advertising and image (40%), followed by a concern to improve society (32%). They also believed that companies were convinced that adopting these types of policies was the way to do better (11%), as can be seen in Figure 8.

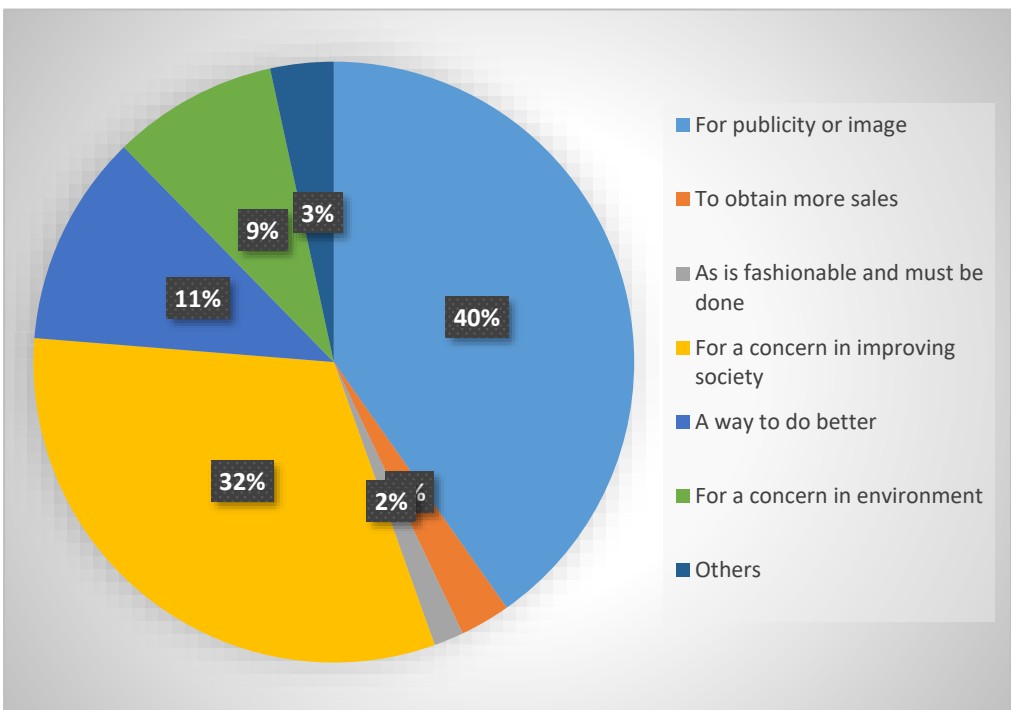

**Figure 8.** Why do you think companies may be interested in adopting Corporate Social Responsibility or CSR policies and practices? Source: Own elaboration.

### 3.3. Qualitative Analysis of the Interviews

In this part, where we analyze the speeches of the directors of the Gloria Company, we are aware that they tried to defend their position by claiming that the crisis was caused by the livestock sector, more interested in selling fresh milk than in any other issue. Although there are certain gaps in the story itself, it can be demonstrated that this statement is only half true, by comparing it with our documentary analysis. In the Gloria Company there is no clear social responsibility policy to attend to suppliers, which leads to conflicts about contract management and payments per liter of milk.

The company and representatives consider themselves as the standard-bearers of the fight against malnutrition in Peru. This is a recent action aimed at cleaning their public image, which provides us with an interesting insight: the Peruvian consumer cares about the price but also, even if only minimally, about the quality of the products. It is worth noting that other food companies such as Laive and Alicorp are carrying out similar actions to place this topic on the public agenda, in collaboration with the media.

Finally, analyzing the speeches of the business representatives, it can be noticed that the company never apologized to its consumers. Their statement is rather defensive, as they only used arguments indicating that the information on the label was incorrect. This is an objective and true fact, although it is interesting to highlight that it was the state entity of the Ministry of Health, DIGESA itself,

that granted the permit for its commercialization. There was probably negligence on the part of the authorities. It should also be considered that the problem with this product, that is not milk, has not only affected the Gloria company and "Pura Vida", but also Laive, a Peruvian competitor, and Nestlé, a foreign company that also competes for this market.

During the last years, people from different parts of the world, both during political elections and in the face of market decisions, have shown an enormous level of conformity in relation to voting and purchasing, even developing an acceptance behavior toward people or products denounced for corruption or legally accused, as in the recent case of Lula da Silva or the one that concerns us: the Gloria Company in Peru. In Table 7 we indicate the main aspects discovered in the qualitative methodological approach.

**Table 7.** Social perception, self-perception of the company and socioinstitutional dimensions.

| | Social Perception | Self-Perception of the Company | General Regulations of the Country and Consumer Acceptance of the Company |
|---|---|---|---|
| Corporate Ethical Discomfort/Welfare | Tendency to accept ethical well-being; but absolute acceptance is missing. | Trying to hide the real situation | Many people related to similar cases |
| Moral Agency Theory | High. People in power (politicians and businessmen) think of their own benefits, rather than the common good. | Few companies have and implement their ethical codes. | There are regulations regarding crimes of corruption, bribery, peculation and others. |
| Corporate Social Responsibility | Little information | Voluntary and related to philanthropy. | No regulation, except in the environmental aspect. |
| Business ethics | Undervalued | Undervalued by entrepreneurs and managers. | Not regulated. Totally voluntary. |

Source: own elaboration.

## 4. Discussion

Blanc and Al-Amoudi (2013) [44] have examined how the legislator should intervene in private sector institutions to counterbalance any unfairness that results from the decline of the welfare state; in our research, we have discovered that the welfare state is also the fundamental basis to develop business ethics. Our results run together with previous research, as the one carried out by Hernani and Hamann (2013) [45]. They built a scale that was applied to 506 undergraduate university students at five universities in Peru and concluded that university students have a low perception of the social responsibility activities carried out by SMEs. The Peruvian company we have studied shows, on the one hand, that this low social perception has a real basis, just because of the incorrect behavior of Gloria company, and on the other hand that people in the country are more affected by the price of the products they consume than by the company's responsibility. Other authors, Marquina and Morales (2012) [46], have also contributed to the ongoing debate regarding the importance of corporate social responsibility as an influential factor in socially responsible consumption.

Nevertheless, Schneider (2014) [8] discovered that businesses play an important role in balancing economic and social considerations, despite the level and features of the welfare state; accordingly, we showed that Peruvians are developing a certain concern in relation to business ethics, just by considering the specific case of "Pura Vida". De Geer et al. (2009) [47] offer the vision of a high level welfare state, as the Swedish one, but also report some difficulties in conceiving CSR and business ethics as an arena to be occupied by companies, and not just by public organizations and the government. So, even in this kind of situation, some problems arise in other countries such as Peru. Companies

should embrace a business ethics strategy because of the huge socioeconomic benefits they could obtain and the positive consequences for society as a whole [48]. Our research matches that of Albareda et al. (2008) [49] in that CSR is linked to cultural and political social values such as the welfare state of a country. They studied three countries in Europe (Italy, Norway and the United Kingdom) and observed these kinds of changes.

Joseph Heath (2009) [50] asks whether the stakeholder paradigm represents the most fruitful way of articulating the moral problems that arise in business and suggests that the concept of market failure offers a more satisfactory framework for the articulation of the social responsibilities of businesses. Our research points to the importance of individual responsibility, showing a case of incorrect business behavior and the social reaction in Peru. Indeed, we agree with Heath on the importance of ethical business commitment when there are market failures. We discovered that even in countries with mild welfare states, there is a social trend toward a demand and a concern for correct business behavior.

Our methodological approach has focused on the social perception—both external, on the part of civil society, and internal, on the part of the members of staff inside the company—of CSR; some other interesting methods could be used to assess the weight and importance of CSR in a given country, such as the ones used by [7] to compare Spain and Poland. One of the main goals of responsible practices is reporting to society, through disclosure information about the social, economic and environmental performance of companies [51]. As our research shows, there is still much to do. Tanaka (2015) [52] gave a reporting model for companies in Peru: with accurate measurement, timely reporting to government authorities and the fulfillment of minimum requirements by companies, the environmental issues in Peru could be solved. In a recent study, Ali, Frynas and Mahmood (2017) [53] found that in contrast to developed countries, firms in developing countries perceive a relatively low pressure from the public with regards to CSR disclosure. This is similar to the low pressure from the public found in relation to the consumption of products from companies with a low ethical commitment and low CSR.

## 5. Conclusions

Consumers and customers of an emerging economy such as the Peruvian are showing concern and a growing trend toward ethical and corporate social responsibility issues, as shown by the results obtained both in our documentary analysis and in the data from the survey. Of those surveyed, 56.60% believe that the environment should be an essential conservation goal for companies; 52.50% are concerned about the attention to customer rights; 49.8% want to address aspects related to business ethics; 56.60% are disposed toward the compliance with regulations and laws in Peru. However, despite these results, the levels of other consolidated socioeconomic environments need to be reached. In general, their perception is limited and diffuse and they are not well informed, nor need to be, about the products and services and their quality. The main motivation in deciding the purchase is the price, which is logical given the subsistence economy and the absence of a welfare state and the equal opportunities that can be found in other developed countries. Therefore, most companies do not care about quality or ethical practices with regard to their various interest groups, and particularly with regard to the customer or final consumer.

In a basically informal economy, where the law of the "smartest" prevails in people's daily behaviors, it is interesting to note that companies transfer this behavior to their daily practices, where what matters is to comply with the norm or the law. However, in some circumstances, they are also tempted not to comply.

The Gloria company, but also its competitors and other Peruvian and foreign companies, have shown that it is easy to deceive the consuming public. It should also be noted that the company never apologized to the consumer, on the grounds that there was an error on the label. However, the most important thing is that the consumer easily forgave the company and the brand and continued buying from them after the crisis. This same fact can be compared with other social dynamics in which citizens forget certain broken promises, for example those of political candidates, and vote again for them even though they have not been fulfilled.

The Gloria company has capitalized on being a leading company and a "lovemark" (Roberts, 2005) [54]. "Lovemark" is a term referring to the fact that consumers and even workers manage to care for a brand. It is a strong feeling, that is achieved when the brand (product or service) becomes an important part of the consumers' life. And this has allowed it to recover from the crisis quickly, although its attractive communication strategy to clean its image was not the best, since it was not credible. In this case, the most valuable assets were the product knowledge and the price. The starting point in business ethics is the concern and personal decision of the founders and leaders. Being ethical is not a business challenge, nor does it entail a public recognition: it is the decision of the people who run the companies. There are ethical companies run by ethical people, regardless of their context.

Our research has limitations that point the way to further studies, mainly because most of the existing literature and research explains only one direction—the one which mainly shows that CSR is important to cover the deficiencies of the welfare state. Our investigation indicates that the welfare state is essential for CSR practices to be carried out, so more and extended research must be done in this idea.

The commitment to business ethics is a pending issue that has to be revalued in emerging societies such as Peru. Committing to, and investing, in ethics and corporate social responsibility is a value that generates long-term sustainability, since it engenders good relationships with the various interest groups. It is worth mentioning that nowadays the most relevant thing for companies is to achieve sustainability and to remain in the market. This is how corporate, social and ethical values are achieved, in a win-win socioeconomic game.

Future lines of research may analyze the degree of social and ethical commitment of companies and other kinds of public organizations in a dynamic and emerging economy such as Peru. The challenge is to build a social welfare state that allows for a socioeconomic transition toward egalitarian and innovative values, for a broad social majority.

**Author Contributions:** R.P.-G., M.V.S.-F. and J.R.-L. have pointed out the research objectives and the theoretical framework; J.R.-L. has proposed the idea and the design of the research, looking for the people who could help to carry out the research; M.V.S.-F. has developed the whole structure of the paper and the conceptualization of the study; R.P.-G. has worked with the statistical package and with all the figures, data analysis, discussion and conclusions. All authors have read and agreed to the published version of the manuscript.

**Funding:** This research received no external funding.

**Conflicts of Interest:** The authors declare no conflict of interest.

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
