# Peer review of "Business Ethics Crisis and Social Sustainability. The Case of the Product “Pura Vida” in Peru"

_sustainability, doi:10.3390/su12083348_

Round 1
Reviewer 1 Report
Overall: an original and important question (about CSR and perceptions and opinions about CSR in developing economies). The impact of culture and different political economies on what we believe and how we even speak about CSR is a fascinating and important topic.
It seems the methodology is sound.
On to suggested edits:
- first, the discussion of conclusions (alternative interpretations and limitations of the study and strength of conclusions) could be expanded.
- next, in either the paragraph starting at line 48 (or the next paragraph), I would recommend citing to work to bolster/support the point that the impacts of culture and political economies on what societies believe, expect, and say about CSR has been identified as important and cite to: Sulkowski, A. J., Parashar, S.P. & Wei, L. (2008). Corporate Responsibility Reporting in China, India, Japan, and the West: One Mantra Does Not Fit All. New England Law Review, 42 (4), 787-808 (finding that there are differences in how businesspeople speak about CSR across these countries/regions).
- you might want to further elaborate on your term "Corporate Ethical Discomfort/Welfare" mentioned in line 45
- at line 61, I would also elaborate what you meant by "first consequence" and social acceptance, as this was not 100% clear what you meant.
- the formatting of Table 5 looked scrambled (I would make sure that it is legible in the final print version).
- also, for attribution of the tables, etc., is "Own Elaboration" the best term to use?
- line 356: the text at this point is not completely understandable - I would revise this section to be more comprehensible.
- line 466: "love mark" - I think you meant to say something about the trademark or brand being well-loved or well-known and respected? I would clarify this.
- the concluding paragraph, starting with line 479, I would also rewrite - it is not clear what you want to communicate.
Again, this article deals wih a vital question, and the research methodology is appropriate, and the context is very interesting.
Most of these suggested changes are edits to the writing, and are not "deep" revisions or critiques related to the study, methods, or findings.
Best of luck, and I hope to read a revision soon. Thank you for the chance to read and review this.
Author Response
- first, the discussion of conclusions (alternative interpretations and limitations of the study and strength of conclusions) could be expanded.
We have expanded the summary, limitations, new research lines and conclusions
- next, in either the paragraph starting at line 48 (or the next paragraph), I would recommend citing to work to bolster/support the point that the impacts of culture and political economies on what societies believe, expect, and say about CSR has been identified as important and cite to:
Sulkowski, A. J., Parashar, S.P. & Wei, L. (2008). Corporate Responsibility Reporting in China, India, Japan, and the West: One Mantra Does Not Fit All. New England Law Review, 42 (4), 787-808 (finding that there are differences in how business people speak about CSR across these countries/regions).
Regarding this point, we have made a deeper bibliographic review including the cited paper and some other more in relation to the main topic of our research. It appears in the last version in the second paragraph after line 48.
- you might want to further elaborate on your term "Corporate Ethical Discomfort/Welfare" mentioned in line 45.
We have explained this new term in the text line 45 and the following.
- at line 61, I would also elaborate what you meant by "first consequence" and social acceptance, as this was not 100% clear what you meant.
This is explained in line 66 & 67.
- the formatting of Table 5 looked scrambled (I would make sure that it is legible in the final print version).
Yes, table 5 is correct in the final version of the paper.
- also, for attribution of the tables, etc., is "Own Elaboration" the best term to use?
Regarding this point, we have been exploring different published papers and this expression appears in most of them, so we consider that this term is correct.
- line 356: the text at this point is not completely understandable - I would revise this section to be more comprehensible.
Yes, we have given a clear explanation about that, now is in line 374 and following.
- line 466: "lovemark" - I think you meant to say something about the trademark or brand being well-loved or well-known and respected? I would clarify this.
We have explained, in the text of our paper, the meaning of lovemark following Kevin Roberts that in 2005 created this term.
- the concluding paragraph, starting with line 479, I would also rewrite - it is not clear what you want to communicate.
We have modified this entire paragraph giving a clear idea of we wanted to say in relation to the future research lines.

Reviewer 2 Report
Authors should make a broader literature review, in particular comparing the degree of social responsibility between different countries. The analysis presented in this paper is interesting, but there is no point of reference.
Second, references should be significantly extended, which is partly connected with previous comments. Describing more papers in this topic would really add value to this paper. Authors could mention in the paper (preferably in literature review) that there are other articles in which different approach or methodology was elaborated. For instance - Garstecki, D., Kowalczyk, M., Kwiecińska, K., CSR Practices in Polish and Spanish Stock Listed Companies: A Comparative Analysis.
Third, authors of this paper should indicate in the summary the possibilities to use and extend this study in their future research activities.
I am convinced that if only authors take into account all my comments, the article will have higher value in represented area of research.
Author Response
Authors should make a broader literature review, in particular comparing the degree of social responsibility between different countries. The analysis presented in this paper is interesting, but there is no point of reference. Second, references should be significantly extended, which is partly connected with previous comments. Describing more papers in this topic would really add value to this paper.
We have made a broader literature review and put other 24 more references inside the text. Some of the papers show different situations among countries.
Authors could mention in the paper (preferably in literature review) that there are other articles in which different approach or methodology was elaborated. For instance - Garstecki, D., Kowalczyk, M., Kwiecińska, K., CSR Practices in Polish and Spanish Stock Listed Companies: A Comparative Analysis.
We have read and reference this interesting paper, mainly because the methodological approach, as reviewer 2 suggested.
Third, authors of this paper should indicate in the summary the possibilities to use and extend this study in their future research activities.
We want to follow this research line open in Peru, applying the methodology to other emergent economies and also trying to show the importance of an ethical business leadership for both the companies and the society.
We have justified the need of more research in this topic and that we will go on through this.
Round 2
Reviewer 2 Report
I have only one request - in references there should be "Garstecki" instead of "Garstcki". This is the wrong fragment:
Garstcki, D.; Kowalczyk, M.; Kwiecińska, K. CSR Practices in Polish and Spanish Stock Listed Companies: A Comparative Analysis, Sustainability. 2019, vol. 11, no 4, p. 1054.
Overall, I find this paper interesting and ready to be published.
Author Response
Dear reviewer: sorry for the mistake in the name of Garstecki, it is already corrected.
Thank you for your comments and support.
Best regards,
The authors